# Increased Prevalence of Psychosocial, Behavioral, and Socio-Environmental Risk Factors among Overweight and Obese Youths in Mexico and the United States

**DOI:** 10.3390/ijerph16091534

**Published:** 2019-04-30

**Authors:** Yvonne N. Flores, Zuelma A. Contreras, Paula Ramírez-Palacios, Leo S. Morales, Todd C. Edwards, Katia Gallegos-Carrillo, Jorge Salmerón, Cathy M. Lang, Noémie Sportiche, Donald L. Patrick

**Affiliations:** 1Unidad de Investigación Epidemiológica y en Servicios de Salud, Delegación Morelos, Instituto Mexicano del Seguro Social, Cuernavaca, Morelos 62000, Mexico; ynflores@ucla.edu (Y.N.F.); kgallegosc13@gmail.com (K.G.-C.); nsportiche@gmail.com (N.S.); 2UCLA Department of Health Policy and Management, Center for Cancer Prevention and Control Research, Fielding School of Public Health and Jonsson Comprehensive Cancer Center, Los Angeles, CA 90095, USA; 3UCLA Department of Epidemiology, Fielding School of Public Health, Los Angeles, CA 90095, USA; zuelmaarellano@ucla.edu; 4Department of Health Services, School of Public Health, University of Washington, Seattle, WA 98195, USA; lsm2010@uw.edu (L.S.M.); toddce.uw@gmail.com (T.C.E.); donald@uw.edu (D.L.P.); 5Universidad Nacional Autónoma de México, Centro de Investigación en Políticas, Población y Salud. Ciudad Universitaria, Ciudad de México 04510, Mexico; jorge.salmec@gmail.com; 6UCLA Department of Community Health Sciences, Fielding School of Public Health, Los Angeles, CA 90095, USA; clang@ucla.edu

**Keywords:** obesity, quality of life, adolescent, risk factors, psychosocial, socio-environmental, behavior, United States, Mexico, Latinos

## Abstract

The aim of this study was to examine various psychosocial, behavioral, and socio-environmental factors in a multiethnic sample of healthy-weight, overweight, and obese youths in the United States (US) and Mexico and determine differences by sex. We conducted a cross-sectional analysis of 633 youths aged 11–18 years who completed a self-reported questionnaire. Height and weight were measured to determine body mass index (BMI). Overweight and obese youth in both countries were significantly more likely to report a higher body image dissatisfaction (Odds Ratio (OR) = 1.67 and OR= 2.95, respectively), depressive symptoms (OR = 1.08 and OR = 1.12, respectively), perceive themselves as overweight (OR = 2.57) or obese (OR = 5.30), and a lower weight-specific quality of life (OR = 0.97 and OR = 0.95, respectively) than healthy-weight youth. Obese youth have lower healthy lifestyle priorities (OR = 0.75) and are less likely to be physically active (OR = 0.79) and eat breakfast (OR = 0.47) than healthy-weight youth. Additionally, overweight and obese youth are more likely to engage in weight control behaviors (OR = 5.19 and OR = 8.88, respectively) and restrained eating than healthy-weight youth. All the aforementioned results had a p-value of <0.05, which was considered statistically significant. The association between these factors and overweight or obesity remained significant after controlling for age, sex, race/ethnicity, and country. In conclusion, obesity was associated with a range of psychosocial, behavioral, and socio-environmental risk factors in both countries. Our findings support the need for multifactorial approaches when developing interventions to address the growing problem of obesity among youth in the US and Mexico.

## 1. Introduction

The high prevalence of obesity among youth is one of the most concerning public health issues in both developed and developing countries [1]. Most overweight or obese children live in developing countries, where the rate of increase is over 30% higher than in more developed countries [1]. In the past 40 years, the number of obese children and adolescents (aged five to 19 years) has increased from 11 million in 1975 to 124 million in 2016 [2]. If current trends continue, by 2022, there will be more obese children and adolescents worldwide than moderately or severely underweight children [2]. Childhood and adolescent overweight and obesity are associated with increased risk of subsequent diabetes, stroke, coronary heart disease, hypertension, functional disability, as well as premature adult mortality and morbidity [3,4]. In the United States (US), an estimated 34.5% of adolescents aged 12–19 years were overweight or obese from 2011–2012, and of these, 16.9% were obese [5]. This number rose to 20.6% in 2015–2016 [6]. Disparities in obesity and overweight exist across racial and ethnic groups in the US, with African American and Mexican-American adolescents ranking highest in prevalence [7,8]. During 2011–2012, the prevalence of overweight and obesity was 39.8% and 38.1% among African American and Latino adolescents, respectively, followed by non-Latino white (31.2%) and Asian (24.6%) adolescents [5]. From 2013–2016, the prevalence of obesity among youth of Mexican origin aged 12–19 years was 26.6%, as compared to 17.2% among non-Latino whites [9]. Studies have also shown that US-born Mexicans are significantly more likely to be overweight or obese than Mexican immigrants [10,11]. In Mexico, the prevalence of obesity and overweight among adolescents aged 12–19 years was 36.3% in 2016 [12].

Addressing obesity is complex, due to its multi-causal nature that includes various psychosocial, behavioral, and socio-environmental factors. Previous studies have found an association between a range of psychosocial factors and increased obesity risk, such as body size dissatisfaction and self-perception of overweight, because they may promote unhealthy weight control behaviors [13,14,15]. Other psychosocial factors that have been examined include exposure to adverse life events and the influence of the family and peer environment, which may be associated with a greater risk of childhood overweight/obesity [16]. Studies have also found that depressive symptoms are a risk factor for obesity because binge eating may be used as a coping mechanism [14,17]. Obese youth report having a lower quality of life (QOL) [18,19,20], which improves when they lose weight [21].

Unhealthy weight control and dietary restraint behaviors have been found to predict the onset of obesity [14,15,17,22]. Studies also show that prioritizing healthy eating may protect youth from becoming overweight or obese [23,24], whereas prioritizing physical activity appears to be less protective [23]. Various studies have demonstrated a negative association between breakfast consumption and an increase in body mass index (BMI) [14,22,25,26,27], which could be due to its association with favorable nutrient intake, improved food choices, and higher physical activity levels [26,28]. The protective effect of physical activity has also been observed in both cross-sectional and longitudinal studies [14,22]. However, the relationship between fast food consumption and obesity has not been established conclusively in the literature. Some studies have shown an inverse association between fast food consumption and obesity [14,27], while others report that fast food consumption is a predictor of weight gain [28,29].

Several socio-environmental factors have also been associated with risk of obesity in adolescents. Parents who have unhealthy lifestyles are more likely to have children who become overweight or obese [30,31,32]. Conversely, positive parental influence regarding healthy diet and frequent physical activity have been associated with reductions in BMI among overweight and obese adolescents [33]. Studies also report that increased availability of healthy food at home encourages healthy eating in adolescents and is protective against overweight and obesity [34,35], while parental obesity is associated with an increased risk of adolescent and ensuing adult obesity [17]. However, Haines et al. found that the availability of healthy food at home and perceived parental obesity did not predict onset of obesity [14]. Parental concern regarding their child’s weight has been positively associated with their child being overweight or obese [14,36]. Parental concern may lead to parental pressure to lose weight and encouragement of restrictive feeding practices, which could lead to weight gain [14,36]. However, parents who reported being concerned about their child’s weight were more likely to improve the family’s diet, limit child screen time, and attempt to increase their child’s physical activity levels [36].

Although there is no individual factor that causes obesity, most research to date has lacked an integrated approach to examine the factors that may be contributing to the high rates of overweight and obesity among youth [17]. An exception would be a study by Haines et al., which looked at the effects of personal, behavioral, and socio-environmental factors on risk of overweight in an ethnically diverse population in Minnesota [14]. To the best of our knowledge, the present study is the first to compare the effects of multiple domains on overweight or obesity risk among a bi-national, ethnically diverse sample of youth. The objective of this study was to identify risk and protective factors for overweight or obesity within the following three domains: Psychosocial, behavioral, and socio-environmental, in a sample of African American, Caucasian, and Latino youths in the US, and Mexican youths in Mexico, and determine differences by sex.

## 2. Research Methods and Procedures

### 2.1. Study Population and Data Collection Procedures

US participants were recruited from community centers, schools, clinics, and youth programs in Seattle, Washington and Los Angeles, California (*n* = 452). A convenience sample of youth was also recruited from the main Mexican Institute of Social Security (IMSS, as per its Spanish abbreviation) hospital in Cuernavaca, Morelos (*n* = 181). Study flyers were posted in various areas of the IMSS clinic, and potential participants were also informed of the study by staff during their visit to the primary care clinics. All individuals who expressed an interest in the study were contacted by a study recruiter who conducted a telephone interview with the primary caregivers of the potential participants to determine eligibility. Participants had to be African American, Caucasian, or Latino, and between the age of 11–18 years. Youths who met study inclusion criteria of age, 5th grade reading ability, and no serious physical or mental illness diagnosis were informed that participation in the study would involve completing a 40-min questionnaire and having their weight, height, and waist circumference measured. All study participants were enrolled between 2006 and 2008, and informed consent was obtained from each participant and a parent or guardian prior to their inclusion in the study. Further details regarding study design, methodology, and baseline participant characteristics are specified elsewhere [19,37,38]. The Institutional Review Boards of the University of Washington, the University of California, Los Angeles, and the Mexican Institute of Social Security approved all study materials including the study questionnaire, protocol, and consent forms (Seattle Children’s Hospital IRB approval number: 11916; IMSS IRB approval number: R-2007-1701-13; UCLA IRB approval number: G06-09-094-01).

Study participants completed a self-administered questionnaire that included the 21-item youth quality of life weight-specific measure (YQOL-W), a generic youth quality of life Instrument (YQOL-R), as well as measures of perceived general health, physical function, body shape satisfaction, and symptoms of depression. The Spanish versions of these measures have been used extensively and validated in other research studies [21,38,39,40,41,42,43,44]. All study materials were designed to be readable and understandable at a 5th grade level.

### 2.2. Study Measures

The following study variables are all reported as indices except for physical activity, which was measured with a single item. Each index score was derived by summing the individual item scores and dividing by the number of items, with the exception of specific scales that have been established and validated (e.g., the YQOL-W, CDI-S, and DEBQ) [19,21,39,45]. The Cronbach’s alpha value indicates the internal consistency of each index that was created.

#### 2.2.1. Psychosocial Factors

Body image dissatisfaction. The body image satisfaction scale consists of the following three questions that ask youth to rate their satisfaction with their weight, body shape, and muscle size [46]: How satisfied are you with your weight?; How satisfied are you with your body shape?; How satisfied are you with your muscle size? A body image dissatisfaction index was created using these questions, which ranges from 1 to 5, with higher values indicating greater dissatisfaction (Cronbach’s α = 0.84).

Depression symptoms. The children’s depression inventory: short version (CDI-S) was used to assess depressive symptoms [39]. The CDI-S consists of 10 items with a total score that ranges from 0 to 20. Higher scores indicate a greater presence of depressive symptoms [39].

Self-perception regarding weight. This index comprises three items with a total score ranging from 1 to 5 (Cronbach’s α = 0.77). A higher score indicates a greater likelihood to regard oneself as overweight or obese. One of the three items is the pictorial body image assessment (PBIA), which asks youth: Which figure in A (female figures) or B (male figures) above is closest to your usual weight? [38] The PBIA silhouettes were modified from Stunkard et al. [47] to include larger body shapes. The silhouettes range from underweight (BMI < 19) to highly severe obesity (BMI > 50) [48]. The two other items ask youth to describe their weight (How do you describe your weight?) [49] and if they ever feel fat (Do you ever feel fat?) [50].

Youth weight-specific quality of life. Weight-specific QOL was evaluated using the YQOL-W, a 21-item instrument with three domain scores: Self, social, and environmental (Cronbach’s α for selfitems = 0.90; Cronbach’s α for social items = 0.90; Cronbach’s α for environmental items = 0.90). The validity and reliability of this instrument has been tested in a multicultural sample of overweight and obese youth in the US and Mexico [19,21]. The YQOL-W has good reliability and validity for assessing weight-specific QOL in children and adolescents, including one-week test–retest intra-class correlation coefficients that were 0.73 for social, 0.71 for self, and 0.73 for environment [19]. The total score for this survey ranges from 0–100, and 100 indicates the best QOL.

#### 2.2.2. Behavioral Factors

Healthy lifestyle priorities. Youth were asked to indicate how much they care about (1) eating healthy food and (2) staying fit and exercising [34]. The following two items: How much do you care about eating healthy food? and How much do you care about staying fit and exercising? were used to construct an index that ranges from 1 to 4, with 4 being the highest level of interest in maintaining a healthy lifestyle (Cronbach’s α = 0.59).

Physically active. Physical activity was assessed with a single item that asks youth to rate their physical activity level compared to others their age. Compared with most boys/girls your age, would you say that you are: (3) More active, (2) less active, (1) about the same, (0) not sure? The response for this item is on a 0–3 point scale, with a higher score indicating that youth consider themselves to be more physically active than their peers [51].

Fast food consumption. An index was constructed using three items: During the past 7 days, how many times did you eat French fries, sweets, chips, or other foods sometimes called “junk food”? How many times in the past 7 days did you eat breakfast, lunch, or dinner from a “fast food” restaurant? and We have “junk food” in my home. A 1 to 5 scale was used, with higher scores suggesting a higher consumption of fast foods (Cronbach’s α = 0.73) [34].

Eats breakfast. A breakfast index was created using two items to classify respondents as having eaten breakfast or not based on a yes–no response (1 = no, 2 = yes): Did you eat breakfast today? and I ate breakfast at home. (Cronbach’s α = 0.58) [34,52].

Weight control behaviors. Participants were asked if they engaged in any of the following weight control behaviors in the past 30 days to lose or keep from gaining weight: (1) Exercise, (2) eating less food, fewer calories, or low-fat foods, (3) fasting, (4) taking diet pills, powders, or liquids [49,52]. A weight control index was created using the following four items based on yes–no responses (1 = no, 2 = yes): During the past 30 days, did you exercise to lose weight or keep from gaining weight?; During the past 30 days, did you eat less food, fewer calories, or foods low in fat to lose weight or keep from gaining weight?; During the past 30 days, did you go without eating for 24 h or more (also called fasting) to lose weight or to keep from gaining weight?; and During the past 30 days, did you take any diet pills, powders, or liquids without a doctor’s advice to lose weight or to keep from gaining weight? (Cronbach’s α = 0.37).

Restrained eating behaviors. Ten items from the Dutch eating behavior questionnaire (DEBQ) were used to assess restrained eating behaviors [45]. The index score ranges from 1 to 5, with 5 indicating a higher frequency of restrained eating practices.

#### 2.2.3. Socio-Environmental Factors

Perceived parental concern regarding adolescent weight. Youths’ perception of their parents’ concern regarding their weight and if they are getting sufficient physical activity was assessed with a two-item index on a 5-point scale, where 5 is the highest level of perceived parental concern. The following two items were used: How concerned are your parents about you becoming overweight? and How concerned are your parents about you not getting enough physical activity? (Cronbach’s α = 0.75) [34].

Perceived parent body size. Two indices that represent how youth perceive their parents’ body size were constructed using images of the PBIA, one for males and the other for females. Participants were asked to select the figure that is closest to the usual adult weight of their mother and father: Which number under the figures in the figure Box A is closest to the usual adult weight of your mother? and Which number under the figures in the figure Box B is closest to the usual adult weight of your father? The PBIA silhouettes were modified from Stunkard et al. [47] to include larger body shapes. The silhouettes range from underweight (BMI <19) to highly severe obesity (BMI >50) [48]. The 13-point response scale for each item depicts a spectrum of silhouettes with 1 representing underweight, and 13 representing extremely obese (Cronbach’s α = 0.47) [38].

Mother/father healthy values. Perception of parent healthy values was evaluated using two items that ask how concerned your parents are about (1) staying fit and exercising and (2) losing weight or preventing weight gain: How much does your mother/father feel about staying fit and exercising (for herself/himself)? and How much does your mother feel about losing weight or keeping from gaining weight (for herself/himself)? Separate indices were constructed for mother and father health values, with each consisting of these two items (Cronbach’s α = 0.73, 0.83, respectively). The response scale for these indices ranges from 1 to 4, with 4 as the highest level of concern [34].

Home availability of healthy foods. An index of fruit and vegetable availability in the home was created using two items that range from 1 to 4, with 4 indicating the highest frequency of healthy food availability in the home: Fruits and vegetables are available in my home … (1) Never, (2) Sometimes, (3) Usually, (4) Always, and Vegetables are served at dinner in my home … (1) Never, (2) Sometimes, (3) Usually, (4) Always. (Cronbach’s α =0.69) [34].

#### 2.2.4. Body Mass Index (BMI)

Height, weight, and waist circumference were measured by trained study staff. Participants were weighed to the nearest 0.1 kg while wearing minimal clothing using a calibrated electronic TANITA scale (model BC533; Tokyo, Japan). Height was determined to the nearest 0.1 cm using a conventional stadiometer, with the youth standing barefoot, with their shoulders in a normal position. BMI was determined to categorize participants as healthy-weight, overweight, or obese, based on the World Health Organization (WHO) age- and sex-specific classifications for youth aged 5 to 19 years [53].

### 2.3. Statistical Analysis

A descriptive analysis of various sociodemographic variables was conducted for the total study population by country of residence and BMI status. Psychosocial, behavioral, and socio-environmental factors were also examined by country of residence and BMI status. Differences between proportions were assessed using chi-square tests of homogeneity, and t-tests were used to calculate differences between means. Test for trend *p*-values were calculated to determine whether there was a linear association between the study variables and BMI status. Odds ratios and 95% confidence intervals for the association between psychosocial, behavioral, and socio-environmental factors and being overweight or obese were calculated using multinomial logistic regression. These results were adjusted for sex, age, race/ethnicity, and country of residence. Standardized odds ratios were determined to facilitate comparisons of the study variables since their score range varied considerably. Standardized odds ratios improve comparison and interpretability of the logistic regression results. Multinomial logistic regression models for males and females were also used to examine any differences by sex, after adjusting for age, race/ethnicity, and country. All *p*-values presented are 2-tailed and a *p*-value of <0.05 was considered statistically significant. All statistical analyses were performed using STATA software, version 12.0 (StataCorp LP, College Station, TX, USA).

## 3. Results

The sociodemographic characteristics of the study sample are compared by BMI status in Table 1. Of the 633 participants, 54% are 11–14 years of age, 46% are between 15 and 18 years old, 52% are female, 22% are African American, 25% are Caucasian, 24% are US Latinos, and 29% are youth who live in Mexico. Thirty percent of youth have a healthy BMI, 30% are overweight, and 40% are obese. Thirty-seven percent of participants are from Seattle, WA, 35% are from Los Angeles, CA, and 29% are from Cuernavaca, Mexico. Chi-square tests were used to assess differences by weight status for each of the study variables, separately by country. There are no significant differences by country of residence in terms of sociodemographic characteristics for each of the three BMI categories, except for education level among the US participants. (Table 1)

Table 2 and Table 3 compare the mean scores for various psychosocial, behavioral, and socio-environmental variables, by country of residence and BMI status. Within the domain of psychosocial factors, overweight or obese youth in Mexico and the US are more likely to report being dissatisfied with their body image, to perceive themselves as overweight or obese, and to have lower weight-specific QOL scores than healthy-weight youth. However, overweight or obese youths in Mexico are not more likely to report more depressive symptoms than healthy-weight youths, unlike obese youths in the US, who are more likely to report depressive symptoms than healthy-weight youths (3.2 vs. 2.3, respectively). The presence of depressive symptoms is greater among healthy-weight, overweight, and obese youth in Mexico (3.1, 3.7, 3.9, respectively) than those in the US (2.1, 2.8, 3.2, respectively); and the weight-related QOL reported by overweight or obese youths in Mexico is lower than those in the US (65.1 and 52.9 vs. 78.1 and 67.0, respectively).

In terms of behavioral factors, obese youth in the US have lower healthy lifestyle priorities (3.0 vs. 3.1, respectively), are less physically active (1.6 vs. 1.8, respectively), and are less likely to eat breakfast (1.5, 1.6, respectively) than healthy-weight youth in the US. Overweight and obese participants in both countries are also significantly more likely to engage in weight control behaviors, such as exercise and restrained eating, as compared to healthy-weight youth. The only statistically significant socio-environmental factors reported by obese youths in both countries include being more likely to think that their parents are concerned about their weight and that their parents have a larger body size than healthy-weight youth. However, overweight youths in the US are not more likely to report that their parents are concerned about their weight, as compared to healthy weight youths (Table 2 and Table 3). All the aforementioned results had a *p*-value of <0.05, which was considered statistically significant.

The standardized and adjusted odds ratios for various psychosocial, behavioral, and socio-environmental factors, by BMI status, among youths in Mexico and the US (controlling for age, sex, race/ethnicity, and country of residence) are reported in Table 4. Overweight and obese youth have significantly greater odds of reporting body image dissatisfaction (OR = 1.67, OR = 2.95), having depressive symptoms (OR = 1.08, OR = 1.12), perceiving themselves as overweight or obese, and having a lower weight-specific QOL (OR = 0.97, OR = 0.95), than healthy-weight youth. Obese youth in both countries also have significantly lower odds of having healthy lifestyle priorities (OR = 0.75), being physically active (OR = 0.79), consuming fast food (OR = 0.68), and eating breakfast (OR = 0.47), than healthy-weight youth. Overweight and obese youth are significantly more likely to engage in weight control behaviors (OR = 5.19, OR = 8.88), such as exercise (OR = 1.99, OR = 2.12), as well as eating less, fewer calories, and lower-fat food (OR = 2.15, OR = 2.32) than healthy-weight youth. In addition, overweight and obese youth have significantly greater odds of restrained eating behaviors (OR = 1.86, OR = 2.35) than healthy-weight youth. Both groups are also significantly more likely to perceive their parent as overweight or obese (OR = 1.49, OR = 1.71), and obese youth have significantly greater odds of reporting that their parents are very concerned about their weight (OR = 1.56), compared to healthy-weight youth. The standardized odds ratio results indicate that the following psychosocial, behavioral, and socio-environmental factors are most significantly associated with overweight and obesity: Perceived body shape (OR = 6.31, OR = 25.89), restrained eating behaviors (OR = 1.7, OR = 2.08), and perceived parent body shape (OR = 1.88, OR = 2.34), respectively. The standardized odds ratios of measures with scales that have a wider range, such as the CDI-S (0–20) and the YQOL-W (0–100), show a stronger association with overweight and obesity, than the non-standardized odds ratios. (Table 4) All the aforementioned results had a *p*-value of <0.05, which was considered statistically significant.

Table 5 presents the logistic regression results for the psychosocial, behavioral, and socio-environmental factors, stratified by sex. Some important differences are observed by sex. For example, overweight or obese boys are more likely to report dissatisfaction with their body image (OR = 1.81 and OR = 3.21, respectively) than girls (OR = 1.59 and OR = 2.78, respectively). However, the presence of depressive symptoms is significantly greater among overweight and obese females (OR = 1.14 and OR = 1.16, respectively) but not among males. Girls are also more likely to perceive themselves as overweight or obese and “feel fat” (OR = 8.91 and OR = 34.28, respectively) than boys (OR = 7.14 and OR = 32.28, respectively). Obese females are significantly less likely to be physically active (OR = 0.72) and eat breakfast than healthy-weight females (OR = 0.40), but this association was not found to be significant among males. Overweight or obese males are more likely to engage in weight control behaviors (OR = 13.77 and OR = 12.69, respectively) than obese females (OR = 8.02), especially exercise (OR = 2.67 and OR = 2.59, respectively) and eating less/few calories/low-fat foods (OR = 3.40 and OR = 2.93, respectively). However, obese girls are significantly more likely to consume diet pills, powders or liquids (OR = 9.59) than boys (Table 5). All the aforementioned results had a *p*-value of <0.05, which was considered statistically significant.

## 4. Discussion

The primary objective of this study was to examine the relevance of various psychosocial, behavioral, and socio-environmental factors among overweight and obese youth in the US and Mexico, and to determine differences by sex. We aimed to address gaps in the current research by studying factors in distinct domains among an ethnically diverse, bi-national sample of youth. Our results support the findings of other studies in the US that have examined similar factors within these three domains [13,14,15,16,17,18,19,20,21,22,23,24,25,26,27,28,29,30,31,32,33,34,35,36]. However, as far as we know, our study is the first to explore the effects of multiple psychosocial factors, behavioral, and socio-environmental factors on overweight or obesity risk in a diverse sample of youth. By simultaneously examining all of these factors in one sample, we were able to contrast the relevance of different risk factors in a single large group, rather than across various studies, which may be difficult to compare. Additionally, to the best of our knowledge, this is one of the first studies to shed light on the association between psychosocial, behavioral, and socio-environmental factors and the presence of overweight or obesity among youth in Mexico and Latinos living in the US.

In our study, psychosocial factors, such as a higher rate of body image dissatisfaction, depressive symptoms, self-perception of overweight, and a lower weight-related QOL, were most strongly associated with overweight or obesity. These results are consistent with other studies, which found a higher prevalence of these factors among overweight or obese youth, as compared to healthy-weight youth [13,14,15,17,19,20]. We found that depressive symptoms are significantly associated with overweight or obesity among girls but not boys. Inconsistent gender differences have previously been reported for the relationship between depressive symptoms and obesity [14,17]. These mixed results could be attributable to variations in study design or assessment of depression [54], or due to the characteristics of the study sample. A meta-analysis of 17 studies concluded that depression is positively associated with BMI but only among females [54]. Interestingly, overweight or obese youths in Mexico did not report more depressive symptoms than healthy-weight youths, unlike obese youths in the US, who did report more depressive symptoms than healthy-weight youths. Our findings also indicate a higher prevalence of depressive symptoms among youths in Mexico than in the US.

In terms of self-perception regarding weight, overweight or obese girls were more likely to perceive themselves as overweight or obese than boys. Similar differences have been observed with adolescent girls being more likely to perceive themselves as overweight or obese than boys [13]. A recent study investigated brain activation using functional magnetic resonance imaging during a body perception task in healthy males and females. They found that images of their own bodies were more salient for the female participants and concluded that females may be more vulnerable than males to conditions involving own body perception [55]. Youths in Mexico reported higher scores for all the “self-perception regarding weight categories”, than youths in the US. Obese adolescents have been shown to report a lower QOL [20], which was also found in this study, with overweight or obese youth reporting significantly lower weight-related QOL than healthy-weight youth. Additionally, the weight-related QOL reported by overweight or obese youths in Mexico was lower than in the US. Notably, self-reported QOL is lower in Mexico than in the US., regardless of weight status.

The multivariate analyses indicate that obese youth were less likely to have healthy lifestyle priorities, be physically active, or eat breakfast. However, when stratified by sex, only obese females were significantly less likely to engage in physical activity. Obese and overweight youth were twice as likely to report that they exercise for weight control, compared to healthy weight youths. There are contradictory findings regarding the effect of physical activity by gender, with one study showing a protective effect only among boys [14] and another only among girls [22]. By contrast, eating breakfast has shown a consistent protective effect for boys and girls in various studies, across different ethnic groups [14,26,27]. Our results also indicate that obese youth are less likely to consume breakfast, but when stratified by sex, this association only remained significant among obese females.

We found that obese males are less likely to report that they eat fast food, as compared to healthy-weight males. Additionally, overweight or obese youths in Mexico are less likely to eat fast food than their counterparts in the US. Previous studies have reported a negative association between eating fast food and obesity among males [27] and females [14,27]. However, other researchers have found that fast food consumption is associated with increased risk of obesity [28,29]. When relying on self-reported behaviors, there may be a higher likelihood of over reporting of socially desirable behaviors, which could explain the inverse association between fast food consumption and obesity observed in this study. Several weight control behaviors were also significantly associated with overweight and obesity in this study. There was a stronger association between weight control behaviors and BMI among males compared to females. Unhealthy weight control behaviors have been shown to predict weight gain in boys and girls [14,15,17,22]. Restrained eating was also found to be a risk factor for obesity in our study, which has previously been reported in other studies [17].

Socio-environmental factors were found to have the least significant associations with overweight or obesity. In this study, obese youth were more likely to believe that their parents are concerned about their weight, which has been previously reported in the literature [14,36]. Parental obesity has also been examined in various studies because children of obese parents may be at greater risk for obesity due to shared genetic and environmental factors [17,56]. In this study, youth who perceived their parents as heavier were more likely to be overweight or obese. Although parental health values and the availability of healthy foods at home have been reported to be significant in other studies [30,31,34,35], no significant associations were found in this study.

This study has some limitations, including that it is cross-sectional, and thus, no conclusions about the direction of causality can be made and there is a possibility of reporting bias. Participants were recruited by means of convenience sampling and might not be representative of their respective weight groups. Additionally, this is an exploratory study with a limited sample size for the participants in Mexico. Future studies should be conducted with a larger sample size that will allow for a higher significance threshold to be set for individual comparisons to compensate for the number of inferences being made. Other limitations include the specific measures that were collected using a self-reported questionnaire, a lack of validated measures, and the fact that some of the behavioral and socio-environmental indices, e.g., “healthy lifestyle priorities,” “physically active,” “mother/father healthy values,” or “home availability of healthy foods”, were created based on a limited number of variables and should be interpreted as preliminary findings. The information provided by the study participants was of a quantitative nature, so we were unable to determine the reason for some of the differences observed by sex or country of origin. Future studies should collect more qualitative data to investigate these differences. A strength of this study is that it explored the issue of overweight and obesity among an ethnically diverse group of youth in the US and Mexico, including African Americans and Latinos, who are disproportionately affected by obesity. Additionally, this study examined a breadth of risk factors that have not been analyzed in a comprehensive and comparative manner. Although some of the indices we created to measure eating behaviors do not have a high reliability score, the associations we observed support the expected relationships, especially when obesity is the main outcome variable. The use of indices in this study to combine various factors also allowed for a robust analysis of complex concepts.

## 5. Conclusions

The results of this bi-national study highlight some of the differences and similarities in various psychosocial, behavioral, and socio-environmental factors among a multiethnic sample of healthy-weight, overweight, and obese youths. We hope our findings help to demonstrate the importance of considering a wide range of risk and protective factors for obesity among adolescents, when planning future studies and interventions. Additionally, our results support the need for multifactorial approaches when developing interventions to address the growing problem of obesity among youth in the US and Mexico. Intervention programs should use an integrated approach that addresses several of these factors to help to reduce the alarmingly high rates of obesity among youth in the US and Mexico. More research is needed on how these factors may interact with each other to cause obesity, since many are interrelated. Our study paves the way for future studies to focus on adopting a transdisciplinary approach to identify and address important risk factors for obesity among youth.

## Figures and Tables

**Table 1 ijerph-16-01534-t001:** Sample characteristics by body mass index (BMI) categories and country (*n* = 633).

Sociodemographic Variables	Mexico (*n* = 181)			*p*-Value ^1^	United States (*n* = 452)			*p*-Value ^1^
	Healthy	Overweight	Obese		Healthy	Overweight	Obese	
	*n* = 43	*n* = 68	*n* = 70		*n* = 143	*n* = 124	*n* = 185	
	*n* (%)	*n* (%)	*n* (%)		*n* (%)	*n* (%)	*n* (%)	
Age (years)								
11–14	20 (46.5)	43 (63.2)	40 (57.1)	0.222	73 (51.1)	67 (54.0)	98 (53.0)	0.883
15–18	23 (53.5)	25 (36.8)	30 (42.9)		70 (49.0)	57 (46.0)	87 (47.0)	
Gender								
Female	19 (44.2)	35 (51.5)	34 (48.6)	0.756	77 (53.9)	66 (53.2)	96 (51.9)	0.936
Male	24 (55.8)	33 (48.5)	36 (51.4)		66 (46.1)	58 (46.8)	89 (48.1)	
Race/ethnicity								
African American	-	-	-	0.451	46 (32.2)	35 (28.2)	59 (31.9)	0.798
Caucasian	-	-	-		53 (37.1)	42 (33.9)	65 (35.1)	
Latino	43 (100.0)	68 (100.0)	70 (100.0)		44 (30.1)	47 (37.9)	61 (33.0)	
Education Level								
Elementary School (≤6th grade)	5 (11.6)	15 (22.1)	14 (20.0)	0.336	24 (16.8)	16 (12.9)	19 (10.3)	**<0.001**
Middle School (7th–9th grade)	18 (41.9)	32 (47.1)	36 (51.4)		28 (19.6)	43 (34.7)	81 (43.58)	
High School (≥10th grade)	20 (46.5)	20 (29.4)	20 (28.6)		41 (28.7)	35 (28.2)	57 (30.8)	
Missing	-	1 (1.5)	-		50 (35.0)	30 (24.2)	28 (15.1)	
Mother’s Education								
Less than High School	26 (60.5)	36 (52.9)	44 (62.9)	0.818	28 (19.6)	17 (13.7)	31 (16.7)	0.886
High School/GED	14 (32.6)	22 (32.4)	17 (24.3)		22 (15.4)	20 (16.1)	37 (20.0)	
Some college	1 (2.3)	2 (2.9)	2 (2.9)		38 (26.6)	42 (33.9)	52 (28.1)	
University or higher	2 (4.7)	7 (10.3)	7 (10.0)		46 (32.2)	39 (31.5)	54 (29.2)	
Don’t know	-	1 (1.5)	-		4 (2.8)	2 (1.6)	3 (1.6)	
Missing	-	-	-		5 (3.5)	4 (3.2)	8 (4.3)	
Father’s Education								
Less than High School	23 (53.5)	34 (50.0)	38 (54.3)	1.124	26 (18.2)	24 (19.4)	34 (18.4)	0.069
High School/GED	13 (30.2)	13 (30.2)	11 (15.7)		28 (19.6)	15 (12.1)	50 (27.0)	
Some college	2 (4.7)	2 (4.7)	8 (11.4)		26 (18.2)	36 (29.0)	30 (16.2)	
University or higher	3 (7.0)	16 (23.5)	9 (12.9)		40 (28.0)	26 (21.0)	43 (23.2)	
Don’t know	-	-	-		14 (9.8)	16 (12.9)	17 (9.2)	
Missing	2 (4.7)	2 (2.9)	4 (5.7)		9 (6.3)	7 (5.7)	11 (6.0)	

Sample sizes may not add up to marginal totals due to missing values. ^1^ Differences between proportions were performed using chi-square tests of homogeneity by weight status for each of the study variables, separately by country. Statistically significant results are in bold.

**Table 2 ijerph-16-01534-t002:** Comparison of various psychosocial, behavioral, and socio-environmental factors by BMI category among youths in Mexico (*n* = 181).

	Range *	Healthy	Overweight	Obese	P_overweight_ ^ⱡ^	P_obese_ ^ⱡ^	P_trend_ ^ƚ^
		Mean ± SD	Mean ± SD	Mean ± SD			
Psychosocial Factors							
Dissatisfied with Body Image	1 to 5	2.7 ± 1.3	**3.1 ± 1.1**	**3.8 ± 0.9**	**0.034**	**<0.001**	**<0.001**
Depression symptoms (CDI-S)	0 to 20	3.1 ± 3.3	3.7 ± 3.2	3.9 ± 3.1	0.364	0.223	0.100
Self-perception regarding weight	1 to 5	2.2 ± 0.8	**2.8 ± 0.7**	**3.5 ± 0.6**	**<0.001**	**<0.001**	**<0.001**
Perceived Body Shape (PBIA)	1 to 13	3.0 ± 1.5	**4.4 ± 1.3**	**6.3 ± 1.6**	**<0.001**	**<0.001**	**<0.001**
Body Weight Description	1 to 5	3.1 ± 0.9	**3.9 ± 0.6**	**4.3 ± 0.6**	**<0.001**	**<0.001**	**<0.001**
Feeling Fat	1 to 5	2.4 ± 1.2	**3.1 ± 1.1**	**3.7 ± 1.0**	**0.006**	**<0.001**	**<0.001**
Youth weight-related quality of life (YQOL-W)	0 to 100	75.8 ± 28.1	**65.1 ± 26.0**	**52.9 ± 26.4**	**0.045**	**<0.001**	**<0.001**
Behavioral Factors							
Healthy lifestyle priorities	1 to 4	2.8 ± 0.6	2.9 ± 0.7	2.8 ± 0.7	0.797	0.723	0.569
Physically active	0 to 3	1.5 ± 1.0	1.6 ± 1.0	1.3 ± 1.0	0.785	0.255	0.176
Fast food consumption	1 to 5	1.1 ± 0.4	1.1 ± 0.5	1.0 ± 0.4	0.765	0.073	0.023
Eats breakfast	1 to 2	1.7 ± 0.4	1.6 ± 0.4	1.6 ± 0.4	0.416	0.205	0.239
Weight control behaviors	1 to 2	1.2 ± 0.2	**1.4 ± 0.2**	**1.4 ± 0.2**	**0.002**	**<0.001**	**0.001**
Exercises	1 to 2	1.5 ± 0.5	**1.8 ± 0.4**	**1.8 ± 0.4**	**0.005**	**0.003**	**0.006**
Eat less, few calories, low-fat foods	1 to 2	1.3 ± 0.5	**1.5 ± 0.5**	**1.6 ± 0.5**	**0.045**	**0.032**	**0.046**
Fasting	1 to 2	1.1 ± 0.3	1.1 ± 0.3	1.2 ± 0.4	0.415	0.174	0.168
Diet pills, powders, or liquids	1 to 2	1.0 ± 0.0	1.0 ± 0.1	1.1 ± 0.2	0.429	0.112	0.058
Restrained eating behaviors (DEBQ-R)	1 to 5	2.1 ± 0.8	**2.6 ± 0.8**	**2.7 ± 0.7**	**0.001**	**<0.001**	**<0.001**
Socio-environmental factors							
Perceived parental concern regarding weight	1 to 5	2.7 ± 1.2	**3.2 ± 1.1**	**3.6 ± 1.0**	**0.0316**	**<0.001**	**<0.001**
Perceived parent body size (PBIA)	1 to 13	4.3 ± 1.2	**4.9 ± 1.5**	**5.6 ± 1.4**	**0.0292**	**<0.001**	**<0.001**
Mother healthy values	1 to 4	2.8 ± 0.7	2.6 ± 0.8	2.6 ± 0.7	0.1260	0.1383	0.178
Father healthy values	1 to 4	2.1 ± 0.6	2.3 ± 0.8	2.3 ± 0.8	0.1656	0.1333	0.191
Home availability of healthy foods	1 to 4	3.1 ± 0.7	3.0 ± 0.7	2.9 ± 0.7	0.5642	0.1474	0.103

* A higher score indicates a greater frequency or agreement; ^¶^ Reference category for comparisons between BMI groups; ^ⱡ^ Differences between means were performed using t-tests; ^ƚ^ Cuzick’s trend test; statistically significant results are in bold.

**Table 3 ijerph-16-01534-t003:** Comparison of various psychosocial, behavioral, and socio-environmental factors by BMI category among youths in the Unites States (*n* = 452).

	Range *	Healthy	Overweight	Obese	P_overweight_ ^ⱡ^	P_obese_ ^ⱡ^	P_trend_ ^ƚ^
		Mean ± SD	Mean ± SD	Mean ± SD			
Psychosocial Factors							
Dissatisfied with Body Image	1 to 5	2.3 ± 0.9	**2.8 ± 1.0**	**3.2 ± 0.9**	**<0.001**	**<0.001**	**<0.001**
Depression symptoms (CDI-S)	0 to 20	2.1 ± 2.6	2.8 ± 3.5	**3.2 ± 3.6**	0.059	**0.001**	**0.003**
Self-perception regarding weight	1 to 5	1.9 ± 0.5	**2.6 ± 0.7**	**3.1 ± 0.7**	**<0.001**	**<0.001**	**<0.001**
Perceived Body Shape (PBIA)	1 to 13	2.4 ± 1.3	**3.9 ± 1.4**	**5.2 ± 1.6**	**<0.001**	**<0.001**	**<0.001**
Body Weight Description	1 to 5	2.8 ± 0.7	**3.5 ± 0.7**	**4.1 ± 0.8**	**<0.001**	**<0.001**	**<0.001**
Feeling Fat	1 to 5	2.0 ± 1.1	**2.8 ± 1.3**	**3.3 ± 1.2**	**<0.001**	**<0.001**	**<0.001**
Youth weight-related quality of life (YQOL-W)	0 to 100	90.4 ± 14.1	**78.1 ± 23.0**	**67.0 ± 27.5**	**<0.001**	**<0.001**	**<0.001**
Behavioral Factors							
Healthy lifestyle priorities	1 to 4	3.1± 0.7	3.0± 0.6	**3.0 ± 0.7**	0.099	**0.042**	**0.028**
Physically active	0 to 3	1.8 ± 1.1	1.6 ± 1.0	**1.6 ± 1.0**	0.105	**0.038**	0.058
Fast food consumption	1 to 5	1.8 ± 0.9	1.8 ± 1.0	**1.6 ± 0.7**	0.639	**0.011**	**0.041**
Eats breakfast	1 to 2	1.6 ± 0.4	1.6 ± 0.4	**1.5 ± 0.4**	0.395	**0.010**	**0.008**
Weight control behaviors	1 to 2	1.3 ± 0.2	**1.3 ± 0.2**	**1.4 ± 0.2**	**0.021**	**0.001**	**<0.001**
Exercises	1 to 2	1.6 ± 0.5	1.7 ± 0.4	**1.7 ± 0.4**	0.053	**0.018**	**0.020**
Eat less, few calories, low-fat foods	1 to 2	1.4 ± 0.5	**1.5 ± 0.5**	**1.6 ± 0.5**	**0.003**	**<0.001**	**0.001**
Fasting	1 to 2	1.1 ± 0.3	1.1 ± 0.3	1.1 ± 0.3	0.577	0.281	0.224
Diet pills, powders, or liquids	1 to 2	1.0 ± 0.2	1.0 ± 0.2	1.1 ± 0.2	0.726	0.691	0.648
Restrained eating behaviors (DEBQ-R)	1 to 5	2.1 ± 0.8	**2.5 ± 0.9**	**2.7 ± 0.8**	**0.001**	**<0.001**	**<0.001**
Socio-environmental factors							
Perceived parental concern regarding weight	1 to 5	2.6 ± 1.3	2.7 ± 1.2	**3.2 ± 1.2**	0.766	**<0.001**	**<0.001**
Perceived parent body size (PBIA)	1 to 13	3.9 ± 1.5	**4.9 ± 1.6**	**5.1 ± 1.5**	**<0.001**	**<0.001**	**<0.001**
Mother healthy values	1 to 4	3.1 ± 0.9	3.1 ± 0.8	3.1 ± 0.8	0.854	0.694	0.992
Father healthy values	1 to 4	2.8 ± 0.9	2.6 ± 1.0	2.9 ± 1.0	0.154	0.255	0.173
Home availability of healthy foods	1 to 4	3.2 ± 0.7	3.2 ± 0.7	3.2 ± 0.8	0.888	0.533	0.391

* A higher score indicates a greater frequency or agreement; ^¶^ Reference category for comparisons between BMI groups; ^ⱡ^ Differences between means were performed using t-tests; ^ƚ^ Cuzick’s trend test; statistically significant results are in bold.

**Table 4 ijerph-16-01534-t004:** Standardized and adjusted odds ratios for psychosocial, behavioral, and socio-environmental factors by BMI status, among youth in Mexico and the US (*n* = 633).

	S_OR	Overweight ^ⱡ^			Obese ^ⱡ^	
		OR (95% CI) ^∞^		S_OR ^¶^	OR (95% CI) ^∞^	
Psychosocial Factors						
Dissatisfied with Body Image	1.74	**1.67 (1.3, 2.1) ***		3.22	**2.95 (2.3, 3.7) ***	
Depression symptoms (CDI-S)	1.29	**1.08 (1.0, 1.2) ****		1.44	**1.12 (1.1, 1.2) ***	
Self-perception regarding weight	4.75	**6.63 (4.3, 10.3) ***		15.13	**27.03 (16.3, 44.8) ***	
Perceived body shape	6.31	**2.57 (2.1, 3.2) ***		25.89	**5.30 (4.1, 6.8) ***	
Body weight description	3.47	**4.02 (2.8, 5.7) ***		10.77	**14.30 (9.4, 21.7) ***	
Feeling fat	2.41	**1.99 (1.6, 2.5) ***		4.29	**3.12 (2.5, 3.9) ***	
Youth weight-related quality of life (YQOL-W)	0.46	**0.97 (0.96, 0.98) ***		0.28	**0.95 (0.94, 0.96) ***	
Behavioral Factors						
Healthy lifestyle priorities	0.87	0.82 (0.6, 1.1)		0.82	**0.75 (0.6, 1.0) ****	
Physically active	0.88	0.88 (0.7, 1.1)		0.79	**0.79 (0.6, 1.0) ****	
Fast food consumption	0.97	0.97 (0.7, 1.3)		0.73	**0.68 (0.5, 0.9) ****	
Eats breakfast	0.87	0.70 (0.4, 1.2)		0.73	**0.47 (0.3, 0.8) ****	
Weight control behaviors	1.46	**5.19 (2.0, 13.2) ***		1.64	**8.88 (3.7, 21.5) ***	
Exercises	1.37	**1.99 (1.3, 3.1) ****		1.41	**2.12 (1.4, 3.2) ***	
Eat less, few calories, low-fat foods	1.47	**2.15 (1.4, 3.3) ***		1.52	**2.32 (1.6, 3.4) ***	
Fasting	1.00	0.99 (0.5, 2.0)		1.18	1.68 (0.9, 3.1)	
Diet pills, powders, or liquids	0.97	0.86 (0.3, 2.6)		1.11	1.64 (0.7, 4.1)	
Restrained eating behaviors (DEBQ-R)	1.70	**1.86 (1.4, 2.4) ***		2.08	**2.35 (1.8, 3.0)***	
Socio-environmental factors						
Perceived parental concern regarding weight	1.15	1.12 (0.9, 1.3)		1.76	**1.56 (1.3, 1.8) ***	
Perceived parent body shape	1.88	**1.49 (1.3, 1.7) ***		2.34	**1.71 (1.5, 2.0) ***	
Mother healthy values	0.94	0.92 (0.7, 1.2)		0.97	0.96 (0.7, 1.2)	
Father healthy values	0.93	0.92 (0.7, 1.2)		1.21	1.22 (1.0, 1.6)	
Home availability of healthy foods	0.94	0.96 (0.7, 1.3)		0.97	0.98 (0.7, 1.3)	

^¶^ Standardized odds ratios; ^ⱡ^ Healthy is reference category for comparison between BMI groups; ^∞^ Adjusted for age, gender, race/ethnicity, and country; * *p*-value ≤ 0.001; ** *p*-value < 0.05; significant results are in bold.

**Table 5 ijerph-16-01534-t005:** Association between psychosocial, behavioral, and socio-environmental factors and overweight or obesity, by sex (*n* = 633).

	Female		Male	
	Overweight ^ⱡ^	Obese ^ⱡ^	Overweight ^ⱡ^	Obese ^ⱡ^
	OR (95% CI) ^∞^	OR (95% CI) ^∞^	OR (95% CI) ^∞^	OR (95% CI) ^∞^
Psychosocial Factors				
Body image dissatisfaction	**1.59 (1.2, 2.2) ****	**2.78 (2.0, 3.8) ***	**1.81 (1.3, 2.5) ***	**3.21 (2.3, 4.5) ***
Depression symptoms (CDI-S)	**1.14 (1.0, 1.3) ****	**1.16 (1.1, 1.3) ***	1.00 (0.9, 1.1)	1.07 (0.97, 1.2)
Self-perception regarding weight	**8.91 (4.5, 17.5) ***	**34.28 (16.1, 73.1) ***	**7.14 (3.7, 13.9) ***	**32.28 (15.1, 69.0) ***
Perceived body shape	**3.43 (2.4, 4.9) ***	**7.07 (4.7, 10.6) ***	**2.16 (1.7, 2.8) ***	**4.45 (3.2, 6.1) ***
Body weight description	**4.45 (2.7, 7.3) ***	**12.94 (7.3, 23.0) ***	**3.87 (2.3, 6.4) ***	**17.37 (9.2, 32.7) ***
Feeling fat	**2.02 (1.5, 2.7) ***	**3.15 (2.3, 4.3) ***	**2.25 (1.6, 3.1) ***	**3.53 (2.5, 4.9) ***
Youth weight-specific quality of life (YQOL-W)	**0.96 (0.95,0.98) ***	**0.95 (0.93, 0.96) ***	**0.98 (0.96, 0.99) ****	**0.96 (0.94, 0.97) ***
Behavioral Factors				
Healthy lifestyle priorities	**0.59 (0.4, 0.9) ****	0.75 (0.5, 1.1)	1.16 (0.8, 1.8)	0.76 (0.5, 1.1)
Physically active	0.76 (0.6, 1.0)	**0.72 (0.5, 0.9) ****	1.01 (0.7, 1.4)	0.87 (0.7, 1.1)
Fast food consumption	1.12 (0.8,1.7)	0.76 (0.5, 1.1)	0.85 (0.6, 1.3)	**0.63 (0.4, 0.9) ****
Eats breakfast	0.51 (0.2, 1.1)	**0.40 (0.2, 0.8) ****	0.90 (0.4, 2.0)	0.50 (0.2, 1.0)
Weight control behaviors	3.00 (0.9, 10.5)	**8.02 (2.4, 26.8) ***	**13.77 (3.2, 59.2) ***	**12.69 (3.3, 49.2) ***
Exercise	1.64 (0.9, 3.0)	**1.88 (1.1, 3.4) ****	**2.67 (1.4, 5.2) ****	**2.59 (1.4, 4.7) ****
Eat less, few calories, low-fat foods	1.53 (0.9, 2.7)	**2.00 (1.2, 3.4) ****	**3.40 (1.8, 6.5) ***	**2.93 (1.6, 5.4) ***
Fasting	0.93 (0.4, 2.4)	1.77 (0.8, 4.0)	1.24 (0.4, 3.9)	1.67 (0.6, 4.6)
Diet pills, powders, or liquids	4.16 (0.4, 38.6)	**9.59 (1.2, 76.7) ****	0.41 (0.1, 2.2)	0.54 (0.1, 2.0)
Restrained eating behaviors (DEBQ-R)	**1.69 (1.2, 2.4) ****	**2.76 (1.9, 4.0) ***	**2.22 (1.5, 3.3) ***	**2.11 (1.5, 3.0) ***
Socio-environmental Factors				
Perceived parental concern regarding weight	1.03 (0.8, 1.3)	**1.50 (1.2, 1.9) ***	1.21 (0.9, 1.5)	**1.61 (1.3, 2.0) ***
Perceived parent body size	**1.47 (1.2, 1.8) ***	**1.67 (1.4, 2.0) ***	**1.56 (1.3, 1.9) ***	**1.83 (1.5, 2.3) ***
Mother healthy values	0.83 (0.6, 1.2)	0.81 (0.6, 1.2)	1.00 (0.7, 1.4)	1.12 (0.8, 1.6)
Father healthy values	0.78 (0.5, 1.1)	1.33 (0.9, 1.9)	1.05 (0.7, 1.6)	1.06 (0.7, 1.5)
Home availability of healthy foods	0.95 (0.6, 1.4)	0.96 (0.7, 1.4)	0.94 (0.6, 1.4)	0.99 (0.7, 1.5)

^ⱡ^ Healthy is reference category for comparison between BMI groups; ^∞^ Adjusted for age, race/ethnicity, and country; * *p*-value ≤ 0.001; ** *p*-value < 0.05; statistically significant results are in bold.

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
