# Peer review of "Increased Prevalence of Psychosocial, Behavioral, and Socio-Environmental Risk Factors among Overweight and Obese Youths in Mexico and the United States"

_ijerph, 2019, doi:10.3390/ijerph16091534_

Round 1

Reviewer 1 Report

l.53: delete: foreign-born

l.61: please fix: youth g report

l.90: please change to: An exception would be a study by Haines et al., which looked

l.92: please change to: To the best of our knowledge , the present study is the first

l.122-123: of depression. The Spanish versions of these measures have been used

l.133-188: when possible please give examples of the items used; also, include Cronbach's alpha whenever possible, for consistency; please do this for all variables you describe in this section.

l.176: please change to: that represent how

Tables 1-5: please rethink your horizontal lines; they are confusing; options (using Table as an example): (1) add a horiz line above the categories (gender, race/ethnicity, etc.); or (2) move your existing horiz lines so they are above the category name (and not below it)

Table 1 & l.211-218: please clarify what you are comparing; I see the p values but am unclear on what groups they are referring to as far as having significant differences; clarify this in l.211-218 and also beneath the table

l.231: this confuses me; wouldn't lower QL be associated with higher depression? they are both indicative of lower quality health

Author Response

Point 1. l.53: delete: foreign-born

Response 1: Thank you very much for reviewing our paper and for your useful suggestions for how to improve our manuscript. As requested, we have deleted the term “foreign-born”. (Line 62)

Point 2. l.61: please fix: youth g report

Response 2: As suggested, we have corrected this mistake and eliminated the “g” from the sentence. (Line 73)

Point 3. l.90: please change to: An exception would be a study by Haines et al., which looked

Response 3: We appreciate this observation and have changed the sentence from “A study by Haines et al. looked” to “An exception would be a study by Haines et al., which looked”, as suggested by the reviewer. (Line 102)

Point 4. l.92: please change to: To the best of our knowledge, the present study is the first

Response 4: As suggested, we have added “the present study” to the sentence, which now indicates: “To the best of our knowledge, the present study is the first to compare…” (Lines 104-105)

Point 5. l.122-123: of depression. The Spanish versions of these measures have been used

Response 5: We appreciate this suggestion and have revised the sentence, which now indicates: “The Spanish versions of these measures have been used extensively and validated in other research studies.”

Point 6. l.133-188: when possible please give examples of the items used; also, include Cronbach's alpha whenever possible, for consistency; please do this for all variables you describe in this section.

Response 6: As suggested, we included examples of the items used, except for the 10-item Children’s Depression Inventory: Short Version (CDI-S), the 21-item YQOL-W instrument, and the 10-item Dutch Eating Behavior Questionnaire (DEBQ). We did not report a Cronbach’s alpha for the CDI-S and DEBQ items, since these are widely used measures that have high Cronbach’s alpha results reported in the literature.

Point 7. l.176: please change to: that represent how

Response 7: We have removed the “s” from represents.

Point 8. Tables 1-5: please rethink your horizontal lines; they are confusing; options (using Table as an example): (1) add a horiz line above the categories (gender, race/ethnicity, etc.); or (2) move your existing horiz lines so they are above the category name (and not below it)

Response 8: As suggested, we moved the existing horizontal lines so they are above the category name on tables 1-5

Point 9. Table 1 & l.211-218: please clarify what you are comparing; I see the p values but am unclear on what groups they are referring to as far as having significant differences; clarify this in l.211-218 and also beneath the table.

Response 9: As requested by the reviewer, we have added the following sentence to the paragraph preceding Table 1: “Chi square tests were used to assess differences by weight status for each of the study variables, separately by country.” We have also included this information at the bottom of Table 1.

Point 10. l.231: this confuses me; wouldn't lower QL be associated with higher depression? they are both indicative of lower quality health.

Response 10: We appreciate the reviewer’s comment and have changed this sentence as follows:  The presence of depressive symptoms is greater among healthy-weight, overweight, and obese youth in Mexico (3.1, 3.7, 3.9, respectively) than in the US (2.1, 2.8, 3.2, respectively); and the weight-related QOL reported by overweight or obese youths in Mexico is lower than in the US. We hope that the revised sentence is less confusing, because we did find that youth in Mexico had more depressive symptoms and lower weight-related quality of life. This is what would be expected, since lower QoL is generally associated with higher rates of depressive symptoms.

Reviewer 2 Report

Per the authors, the aim of this study was to examine various psychosocial, behavioral, and socio-environmental risk factors in a multiethnic sample of healthy weight, overweight, and obese youths ages 11-18  in the United States and Mexico. 

While this is a valuable topic, the authors overstate the novelty of this project when they write in the conclusion “our study is the first to explore  the effects of multiple psychosocial factors, behavioral, and socio-emotional factors on overweight or obesity risk in a diverse sample of youth (lines 298-299).” In truth, many studies have been done and are even referenced by the authors to support their rationale for including study variables. 

This study has a number of significant limitations. First, the Authors make a case for examining between-group differences (by weight group and country) for a large number of variables (22) but they do not adjust power to account for potential Type I errors.  The Authors should set a higher significance threshold for individual comparisons to compensate for the number of inferences being made. Nunnally (1978) recommended sampling at least ten times as many subjects as variables. 

Only three of the thirteen study measures (CDI-S, YQOL-W, and DEBQ) appear to have been previously validated. While it is sometimes necessary to create items for constructs not apparent in the extant literature, it is not appropriate to do so when validated measures already exist. As described by the Authors, the measures created for use in this study are insufficient. More specifically, Healthy Lifestyle Priorities is more than eating healthy food and staying fit and exercising. Staying fit and exercising are not synonymous; someone may hate exercising but want to stay fit through activities of daily living. Drinking water, getting sufficient sleep, managing stress…all of these merit inclusion in a healthy lifestyle scale. The Physically Active scale is problematic in that comparisons of “me” and “others my age” is completely subjective and subject to bias. A better assessment would be “me” compared to recommended benchmarks of activity. Fast Food Consumption is vague. Weight Control Behaviors includes both (potentially) healthy and unhealthy behaviors. Some people may engage in these behaviors as part of a healthy lifestyle rather than an attempt to lose weight but they would be grouped together. Someone who exercises and watches calorie intake could theoretically score the same as someone who fasts and takes diet pills. The Authors need to say more about the pictorial body image assessment (PBIA). Is this a validated measure? Professional anthropometric drawings are necessary to accurately match BMI classification categories as “underweight,” “overweight,” and “obese.” Why do the Authors choose to focus on availability of fruits and vegetables and not availability of less healthy foods in the home? Studies suggest availability of unhealthy foods may be a better predictor of food choice. (e.g., Pechey R, Marteau TM. Availability of healthier vs. less healthy food and food choice: an online experiment. BMC Public Health. 2018;18(1):1296. Published 2018 Nov 29. doi:10.1186/s12889-018-6112-3).  The Authors need to provide additional information about how BMI was collected (by who, wearing light clothes, etc.,). 

There are some confusingly written sentences (e.g., lines 122-124) as well as typographical errors (line 61) that should be corrected. 

The Authors state “interestingly the presence of depressive symptoms is greater among…youth in Mexico…yet weight-related QOL…in Mexico is lower than in the U.S. (lines 229-232). The Authors wrote this sentence in a way that suggests this is a surprising finding when, in fact, the relationship (higher depression, lower QOL) is intuitive and supported by research literature. 

The Authors should attempt to explain the seemingly contradictory finding among youth in the US (Table 3) that obese participants had lower Healthy Lifestyle Priority scores but were more likely to report engaging in Weight Control Behaviors, in particular, Exercise and Eat Less/Few Calories/Low Fat Foods.  There is a similar apparent contradiction in Table 4 where obese participants suggest they are less Physically Active than normal weight peers but they also report engaging in more Exercise as a Weight Control Behavior. (This speaks to the less than ideal operationalization of constructs). 

The Authors suggest a significant association between overweight/obesity and restrained eating behavior (line 268) but “restrained eating” is only one part of the Dutch Eating Behavior Questionnaire. Emotional and External Eating Behavior are also components of the DEBQ. Again, the Authors need to be careful how they discuss constructs. 

The Authors write that “weight-related QOL reported by overweight or obese youths in Mexico was lower than in the US (line 323-324).” It is worth noting that self-reported QOL is lower in Mexico than in the US, regardless of weight status. 

The sentence, “The obese youth in our study were less likely to report that they have healthy lifestyle priorities, are physically active, or eat breakfast (lines 325-326) is  very confusing as written. Data in the table suggests they are more likely to exercise (for weight control), correct?   

The Authors assert, “…obese youth were more likely to have parents who were concerned about their weight… (lines 347-348). This isn’t technically true. Participants reported how much they believed their parents were concerned (subjective) but it wasn’t actually parent report. The references appear to be parent-report. 

In addition to the limitations listed (lines 354-357), the Authors should include lack of validated measures, poorly operationalized constructs, and a relatively small sample size for the number of analyses that were run. 

Author Response

Response to Reviewer 2 Comments

Point 1. Per the authors, the aim of this study was to examine various psychosocial, behavioral, and socio-environmental risk factors in a multiethnic sample of healthy weight, overweight, and obese youths ages 11-18  in the United States and Mexico. 

Response 1: Thank you very much for reviewing our paper and for your kind suggestions.

Point 2. While this is a valuable topic, the authors overstate the novelty of this project when they write in the conclusion “our study is the first to explore  the effects of multiple psychosocial factors, behavioral, and socio-emotional factors on overweight or obesity risk in a diverse sample of youth (lines 298-299).” In truth, many studies have been done and are even referenced by the authors to support their rationale for including study variables. 

Response 2: We agree with the reviewer that this is a valuable topic that has been explored in many studies, however as indicated in the Discussion section, to the best of our knowledge, our study is the first to explore the effects of multiple psychosocial factors, behavioral, and socio-environmental factors on overweight or obesity risk in a diverse sample of youth in the USA and Mexico. By simultaneously examining all of these factors in one sample, we were able to contrast the relevance of different risk factors in a single large group, rather than across various studies, which may be difficult to compare. Although we are not the first researchers to investigate the association between weight and various psychosocial, behavioral, and socio-environmental factors among youth, we are the first study to examine these factors in a sample of Mexican youths. We strongly believe it is important for these results to be reported, especially considering the high rates of obesity and overweight among youths in Mexico (36%).

Point 3. This study has a number of significant limitations. First, the Authors make a case for examining between-group differences (by weight group and country) for a large number of variables (22) but they do not adjust power to account for potential Type I errors.  The Authors should set a higher significance threshold for individual comparisons to compensate for the number of inferences being made. Nunnally (1978) recommended sampling at least ten times as many subjects as variables.

Response 3: We appreciate the reviewer’s suggestion to set a higher significance threshold for individual comparisons to compensate for the number of inferences being made and to use a larger sample size. We have included the following sentence to the Limitations paragraph in our Discussion section, to address the issues raised by this reviewer:

“Additionally, this is an exploratory study with a limited sample size for the participants in Mexico. Future studies should be conducted with a larger sample size that will allow for a higher significance threshold to be set for individual comparisons to compensate for the number of inferences being made.”

Point 4. Only three of the thirteen study measures (CDI-S, YQOL-W, and DEBQ) appear to have been previously validated. While it is sometimes necessary to create items for constructs not apparent in the extant literature, it is not appropriate to do so when validated measures already exist. As described by the Authors, the measures created for use in this study are insufficient. More specifically, Healthy Lifestyle Priorities is more than eating healthy food and staying fit and exercising. Staying fit and exercising are not synonymous; someone may hate exercising but want to stay fit through activities of daily living. Drinking water, getting sufficient sleep, managing stress…all of these merit inclusion in a healthy lifestyle scale. The Physically Active scale is problematic in that comparisons of “me” and “others my age” is completely subjective and subject to bias. A better assessment would be “me” compared to recommended benchmarks of activity. Fast Food Consumption is vague. Weight Control Behaviors includes both (potentially) healthy and unhealthy behaviors. Some people may engage in these behaviors as part of a healthy lifestyle rather than an attempt to lose weight but they would be grouped together. Someone who exercises and watches calorie intake could theoretically score the same as someone who fasts and takes diet pills. The Authors need to say more about the pictorial body image assessment (PBIA). Is this a validated measure? Professional anthropometric drawings are necessary to accurately match BMI classification categories as “underweight,” “overweight,” and “obese.” Why do the Authors choose to focus on availability of fruits and vegetables and not availability of less healthy foods in the home? Studies suggest availability of unhealthy foods may be a better predictor of food choice. (e.g., Pechey R, Marteau TM. Availability of healthier vs. less healthy food and food choice: an online experiment. BMC Public Health. 2018;18(1):1296. Published 2018 Nov 29. doi:10.1186/s12889-018-6112-3).  The Authors need to provide additional information about how BMI was collected (by who, wearing light clothes, etc.,). 

Response 4: We understand the reviewer’s concerns regarding the specific study measures we used and the indices we created based on published results in the literature. Although the psychosocial, behavioral, and socio-environmental factors we examined as part of this study have been investigated in prior studies, few studies have compared the effect of these factors on overweight or obesity risk in a diverse sample of youth in the USA and none have done so with youth in Mexico. In terms of the indices we created for this study, we were limited by the type of data that was collected for this study. For example, the “Healthy Lifestyle Priority” variable was created using the youths’ responses regarding how much they care about (1) eating healthy food, and (2) staying fit and exercising, because those were the items that were available to us. We do not have information about the amount of water they drink, the number of hours they sleep, or how they deal with stress. Thus, the results we present are based on the variables that were available to us. We have now included the following text in the limitations section:  “Another limitation concerns the available variables that were used to create the specific psychosocial, behavioral, and socio-environmental indices we examined. Some of the behavioral and socio-environmental indices, e.g. “Healthy Lifestyle Priorities,” “Physically Active,” “Mother/Father Healthy Values,” or “Home Availability of Healthy Foods,” were created based on a limited number of variables and should be interpreted as preliminary findings.”

In response to the reviewer’s request, we have added the following information to describe the pictorial body image assessment (PBIA) that was used for this study, including the appropriate references. “The PBIA silhouettes were modified from Stunkard et al. [43] to include larger body shapes. The silhouettes range from underweight (BMI < 19) to highly severe obesity (BMI > 50) [44].”

Our response to the reviewer’s question: “Why do the Authors choose to focus on availability of fruits and vegetables and not availability of less healthy foods in the home?” is the same as the explanation we provided above. We focused on the availability of fruits and vegetables for the “Home Availability of Healthy Foods” index because that information was collected as part of this study and available for this analysis. We had accesses to information about “Fast Food Consumption”, which we included as a separate index.

As requested by the reviewer, we have included the following information about how BMI was determined. “Height, weight, and waist circumference were measured by trained study staff. Participants were weighed to the nearest 0.1 kg while wearing minimal clothing using a calibrated electronic TANITA scale (model BC533; Tokyo, Japan). Height was determined to the nearest 0.1 cm using a conventional stadiometer, with the youth standing barefoot, with their shoulders in a normal position.”

Point 5. There are some confusingly written sentences (e.g., lines 122-124) as well as typographical errors (line 61) that should be corrected. 

Response 5: The typographical error found in line 61 has been corrected. As suggested, we have also revised lines 122-124 as follows so they are less confusing: “The Spanish versions of these measures have been used extensively and validated in other research studies [18,34-40]. All study materials were designed to be readable and understandable at a 5th grade level.”

Point 6. The Authors state “interestingly the presence of depressive symptoms is greater among…youth in Mexico…yet weight-related QOL…in Mexico is lower than in the U.S. (lines 229-232). The Authors wrote this sentence in a way that suggests this is a surprising finding when, in fact, the relationship (higher depression, lower QOL) is intuitive and supported by research literature. 

Response 6: We appreciate the suggestion and have revised this sentence in the following way to reflect that higher depression has been associated with a lower QOL: “The presence of depressive symptoms is greater among healthy-weight, overweight, and obese youth in Mexico (3.1, 3.7, 3.9, respectively) than in the US (2.1, 2.8, 3.2, respectively); and the weight-related QOL reported by overweight or obese youths in Mexico is lower than in the US.”

Point 7. The Authors should attempt to explain the seemingly contradictory finding among youth in the US (Table 3) that obese participants had lower Healthy Lifestyle Priority scores but were more likely to report engaging in Weight Control Behaviors, in particular, Exercise and Eat Less/Few Calories/Low Fat Foods.  There is a similar apparent contradiction in Table 4 where obese participants suggest they are less Physically Active than normal weight peers but they also report engaging in more Exercise as a Weight Control Behavior. (This speaks to the less than ideal operationalization of constructs). 

Response 7: We agree with the reviewer that it is important to acknowledge the limitations of some of our constructs. We have added the following sentence to the Limitations paragraph of the Discussion section: “Other limitations include a lack of validated measures and the fact that some of the behavioral and socio-environmental indices, e.g. “Healthy Lifestyle Priorities,” “Physically Active,” “Mother/Father Healthy Values,” or “Home Availability of Healthy Foods,” were created based on a limited number of variables and should be interpreted as preliminary findings.”

Point 8. The Authors suggest a significant association between overweight/obesity and restrained eating behavior (line 268) but “restrained eating” is only one part of the Dutch Eating Behavior Questionnaire. Emotional and External Eating Behavior are also components of the DEBQ. Again, the Authors need to be careful how they discuss constructs. 

Response 8: As indicated previously, we are limited by the availability of the data that were collected as part of this study. Only the “restrained eating” items from the DEBQ were included in the study questionnaire, so we focused on the data we had. In response to the reviewer’s suggestion, we added the following information to the Limitations paragraph of the Discussion section: “Some of the indices were created based on a limited number of variables and should be interpreted as preliminary findings.”

Additionally, we have revised the description of how we constructed the “Restrained Eating Behaviors” variable as follows: “Restrained Eating Behaviors. 10-items from the Dutch Eating Behavior Questionnaire (DEBQ) were used to assess restrained eating behaviors [41].”

Point 9. The Authors write that “weight-related QOL reported by overweight or obese youths in Mexico was lower than in the US (line 323-324).” It is worth noting that self-reported QOL is lower in Mexico than in the US, regardless of weight status. 

Response 9: We appreciate the reviewer’s comment and have included the following sentence: “Notably, self-reported QOL is lower in Mexico than in the US, regardless of weight status.”

Point 10. The sentence, “The obese youth in our study were less likely to report that they have healthy lifestyle priorities, are physically active, or eat breakfast (lines 325-326) is  very confusing as written. Data in the table suggests they are more likely to exercise (for weight control), correct?  

Response 10:  As suggested by the reviewer, we have modified the sentence to make it clearer and easier to understand: “The multivariate analyses indicate that obese youth were less likely to have healthy lifestyle priorities, be physically active, or eat breakfast.” Additionally, we included the following sentence: “Obese and overweight youth were twice as likely to report that they exercise for weight-control, than healthy weight youths.”

Point 11. The Authors assert, “…obese youth were more likely to have parents who were concerned about their weight… (lines 347-348). This isn’t technically true. Participants reported how much they believed their parents were concerned (subjective) but it wasn’t actually parent report. The references appear to be parent-report. 

Response 11:  We appreciate the reviewer’s comment and have revised the sentence as follows: “In this study, obese youth were more likely to believe that their parents are concerned about their weight…”

Point 12. In addition to the limitations listed (lines 354-357), the Authors should include lack of validated measures, poorly operationalized constructs, and a relatively small sample size for the number of analyses that were run

Response 12: As requested by the reviewer, we have added the following text to the Limitations paragraph in the Discussion section: “Additionally, this is an exploratory study with a limited sample size for the participants in Mexico. Future studies should be conducted with a larger sample size that will allow for a higher significance threshold to be set for individual comparisons to compensate for the number of inferences being made. Other limitations include a lack of validated measures and the fact that some of the behavioral and socio-environmental indices, e.g. “Healthy Lifestyle Priorities,” “Physically Active,” “Mother/Father Healthy Values,” or “Home Availability of Healthy Foods,” were created based on a limited number of variables and should be interpreted as preliminary findings.”

Reviewer 3 Report

The article provides information novel about the psychosocial, behavioural, and socio-environmental risk factors in overweight obese youths in Mexico and USA. However, although the sample size is large, I need major revision for understanding the final aim and rationale of the study, as well as the discussion with previous studies.

Abstract

In the results section of the abstract, I would include the p value and data supporting such affirmations or results.

Keywords: Authors should report other keywords different to those who are in the title for spreading out your research.

Introduction

Line 41-42. The affirmation should be supported by a citation

Line 51. There is a typo error, “from 2013-2106” should be “from 2013-2016”

Do the authors have more updated information about the percentage of overweight/obesity? What is expected for the near future? Author should report it. Several organizations use to calculate the prevision of the percentage for the prevalences expected in the following years.

What are the psychosocial factors that increased the obesity risk? Author should state and specify more about such info.

Line 67. Its association of what? Is not clear, although its supposed to be bmi.

Line 75-76. There is an interesting publication in regards to this idea, of how the Parental bmi can influence their children’s behaviors. See PMID (28466851)

Methods

Line 127. About the Physical activity measurements, how extensible is the measure of PA in one question? How valid and reliable is it?

Line 148. How is the validity and reliability?

Do the authors checked the sex-interaction to include males and females together? If so, they should report in the statistical analyses or results section.

Line 222-242:  please provide the data, p value at least

Results

In table 2 and 3, I was expecting a comparison results between both countries. However, the results are shown independently by country. I would suggest to compare the data in Mexico and usa since is the most relevant aims of your study. Another option would be the replication of the same idea with different population in order to corroborate or contrast the findings. However, although the tables seems to show it, the aims and structure of the article is not design for this last aim.

Please, include OR whenever is possible. Lines 253-272, and 277-287.

Is table 5 needed? There is not aim showing the differences between males and females, isn’t it? If you include it, I would specify it as well as the BMI in the aim

Discussion

Line 297. Needs citations to support the affirmation

How the authors think that their article is providing information to the literature published? Not only being the first study examining such associations could be determinant for the interest of the study.

Line 311- Only to the variastion of the study design or assessment of depression are the differences? I would suggest also to the characteristics of the study sample

Did the meta-analysis provide any information or rationale of why overweight-obese females are more depressed? Could be something physiological in females that not occur in males?

318-319. Why do you think that girls are more likely to perceive themselves as overweight or obese? Maybe they are more strict with theirselves?

Based on the interaction analyses, we will delete or justify why some associatons are occurred in girls and not in boys or viceversa.

I would discuss more the findings obtained along the discussion section, why they are agree or not with other studies? What could be the justification of the findings?

In limitations, author should state also the questionnaire and self-reported measures as limitation. Also the differences in number of the sample size (Mexico much lower than usa)

Author Response

Point 1. The article provides information novel about the psychosocial, behavioural, and socio-environmental risk factors in overweight obese youths in Mexico and USA. However, although the sample size is large, I need major revision for understanding the final aim and rationale of the study, as well as the discussion with previous studies.

Response 1: Thank you very much for reviewing our paper and for your kind suggestions. We have revised our paper to include more information about the final aim and rationale of the study, as well as a greater discussion of previous studies. As suggested by the reviewer, we now indicate in the abstract and at the end of our Introduction that our final aim was also to determine differences by sex.

“The aim of this study was to examine various psychosocial, behavioral, and socio-environmental factors in a multi-ethnic sample of healthy-weight, overweight, and obese youths in the United States (US) and Mexico, and determine differences by sex.”

We added additional information to the Discussion section and compare our results to those of 27 other published studies in the literature. Since this was an exploratory study that is based on the available data that was collected we were limited in our ability to explain, justify or determine the reason for many of our findings study.

Point 2. Abstract. In the results section of the abstract, I would include the p value and data supporting such affirmations or results.

Response 2: We agree with the reviewer and have included the ORs for the main results presented in the abstract, and also indicate that “All the aforementioned results had a p-value of <0.05, which was considered statistically significant.” (Lines 28-35)

Point 3. Keywords: Authors should report other keywords different to those who are in the title for spreading out your research.

Response 3: We appreciate the comment and have added the term “risk factors” to our key words. (Line 41) Some of the different terms we are using as key words, which are not in the title include: Quality of Life, adolescent and Latino. If the reviewer has any other suggestions for keywords, we would be happy to include them as well.

Point 4. Introduction. Line 41-42. The affirmation should be supported by a citation

Response 4: As requested by the reviewer, we have added the following reference to support the first sentence in the Introduction (Line 46):

World Health Organization. Facts and Figures on Childhood Obesity. October 2017. Available online: https://www.who.int/end-childhood-obesity/facts/en/

Point 5. Line 51. There is a typo error, “from 2013-2106” should be “from 2013-2016”

Response 5: We appreciate the suggestion and have corrected this typographical error. (Line 59)

Point 6. Do the authors have more updated information about the percentage of overweight/obesity? What is expected for the near future? Author should report it. Several organizations use to calculate the prevision of the percentage for the prevalences expected in the following years.

Response 6: As suggested by the reviewer, we have included more updated information about the global prevalence of overweight/obesity and what is expected for the future. The following sentences were added at the beginning of the introduction: “Most overweight or obese children live in developing countries, where the rate of increase is over 30% higher than in more developed countries [1] In the past 40 years, the number of obese children and adolescents (aged five to 19 years) has increased from 11 million in 1975 to 124 million in 2016 [2] If current trends continue, by 2022 there will be more obese children and adolescents worldwide than moderately or severely underweight [2].” (Lines 46-51)

The prevalence of overweight and obesity reported in the United States and Mexico is the most up-to-date.

Point 7. What are the psychosocial factors that increased the obesity risk? Author should state and specify more about such info.

Response 7: As requested by the reviewer, we have added more information about the psychosocial factors that increase risk of obesity to the second paragraph of the Introduction: “Previous studies have found an association between a range of psychosocial factors and increased obesity risk, such as body size dissatisfaction and self-perception of overweight, because they may promote unhealthy weight control behaviors [13-15]. Other psychosocial factors that have been examined include exposure to adverse life events and the influence of the family and peer environment, which may be associated with a greater risk of childhood overweight/obesity [16]. Studies have also found that depressive symptoms are a risk factor for obesity because binge eating may be used as a coping mechanism [14,17]. Obese youth report having a lower quality of life (QOL) [18-20], which improves when they lose weight [21].” (Lines 65-73)

Point 8. Line 67. Its association of what? Is not clear, although its supposed to be bmi.

Response 8: We appreciate the reviewer’s suggestion and have modified the sentence to make it clearer: “Various studies have demonstrated a negative association between breakfast consumption and an increase in body mass index (BMI)…” (Line 78)

Point 9. Line 75-76. There is an interesting publication in regards to this idea, of how the Parental bmi can influence their children’s behaviors. See PMID (28466851)

Response 9: We thank the reviewer for suggesting this additional reference, which we have included (Line 87):

Cadenas-Sanchez C, Henriksson P, Henriksson H, Delisle Nyström C, Pomeroy J, Ruiz JR, Ortega FB, Löf M. Parental body mass index and its association with body composition, physical fitness and lifestyle factors in their 4-year-old children: results from the MINISTOP trial. Eur J Clin Nutr. 2017 Oct;71(10):1200-1205. doi: 10.1038/ejcn.2017.62. Epub 2017 May 3.

Point 10. Methods.Line 127. About the Physical activity measurements, how extensible is the measure of PA in one question? How valid and reliable is it?

Response 10: We agree with the reviewer that measuring physical activity with only one question is an important limitation of our study. We have revised the limitation paragraph in the Discussion section to indicate that: “Other limitations include the specific measures that were collected using a self-reported questionnaire, a lack of validated measures, and the fact that some of the behavioral and socio-environmental indices, e.g. “Healthy Lifestyle Priorities,” “Physically Active,” “Mother/Father Healthy Values,” or “Home Availability of Healthy Foods,” were created based on a limited number of variables and should be interpreted as preliminary findings.” (Lines 423-427) However, despite this limitation, our multivariate analyses do indicate that obese youth were less likely to be physically active, and obese and overweight youth were twice as likely to report that they exercise for weight-control, than healthy weight youths. Future studies shall carry out a more comprehensive evaluation of physical activity, capturing since type, frequency and intensity. 

Point 11. Line 148. How is the validity and reliability?

Response 11: As suggested by the reviewer, we have included more information about the validity and reliability of the YQOL-W:  “Weight-specific QOL was evaluated using the YQOL-W, a 21-item instrument with three domain scores: Self, Social, and Environmental (Cronbach’s α for Self items=0.90; Cronbach’s α for Social items=0.90; (Cronbach’s α for Environmental items=0.90). The validity and reliability of this instrument has been tested in a multicultural sample of overweight and obese youth in the US and Mexico [19,21]. The YQOL-W has good reliability and validity for assessing weight-specific QOL in children and adolescents, including a one-week test–retest intra-class correlation coefficients were 0.73 for Social, 0.71 for Self, and 0.73 for Environment [19].” (Lines 160-166)

We also provide the following reference, which has more information about the validity and reliability of the YQOL-W measure:

Morales, L.S.; Edwards, T.C.; Flores, Y.; Barr, L.; Patrick, D.L. Measurement properties of a multicultural weight-specific quality-of-life instrument for children and adolescents. Qual Life Res 2011, 20, 215-224, doi:10.1007/s11136-010-9735-0.

Point 12. Do the authors checked the sex-interaction to include males and females together? If so, they should report in the statistical analyses or results section.

Response 12: We are not sure what the reviewer means by “checked the sex-interaction to include males and females together.” However, we do present the results of the association between psychosocial, behavioral, and socio-environmental factors and overweight or obesity, by sex in Table 5. We think it is very important to present these results stratified by sex, so as suggested by the reviewer we now indicate in the abstract and at the end of our Introduction that our final aim was also to determine differences by sex.

Point 13. Line 222-242:  please provide the data, p value at least

Response 13: As suggested by the reviewer, we have included some of the specific results reported in Tables 2 and 3. (Lines 274-280) Since the results reported on Tables 2 and 3 are scores, we think that describing the results is more effective and easy to understand than providing six scores per finding (e.g. healthy weight vs. overweight, vs. obese for both Mexico and the US). However, if the reviewer would like for us to add ALL of the significant results reported on Tables 2 and 3, we would be happy to do so. We have also added the following sentence to indicate that: “All the aforementioned results had a p-value of <0.05, which was considered statistically significant.” (Lines 287-288)

Point 14. Results. In table 2 and 3, I was expecting a comparison results between both countries. However, the results are shown independently by country. I would suggest to compare the data in Mexico and usa since is the most relevant aims of your study. Another option would be the replication of the same idea with different population in order to corroborate or contrast the findings. However, although the tables seems to show it, the aims and structure of the article is not design for this last aim.

Response 14: We appreciate the suggestion and have included more information comparing the results between Mexico (Table 2) and the US (Table 3). (Lines 274-288)

Point 15. Please, include OR whenever is possible. Lines 253-272, and 277-287.

Response 15: As suggested by the reviewer, we have included the specific OR results for Table 4 (Lines 302-313) and Table 5 (Lines 329-340). We have also added the following sentence to indicate that: “All the aforementioned results had a p-value of <0.05, which was considered statistically significant.” (Lines 320-321)

Point 16. Is table 5 needed? There is not aim showing the differences between males and females, isn’t it? If you include it, I would specify it as well as the BMI in the aim

Response 16: We think it is very important to present the results stratified by sex, so as suggested by the reviewer we now indicate in the abstract and at the end of our Introduction that our final aim was also to determine differences by sex: “The objective of this study was to identify risk and protective factors for overweight or obesity within the following three domains: psychosocial, behavioral and socio-environmental, in a sample of African American, Caucasian, and Latino youths in the US, and Mexican youths in Mexico, and determine differences by sex.” (Lines 105-108)

Point 17. Discussion. Line 297. Needs citations to support the affirmation

Response 17: As suggested, we have added the corresponding references to support the following sentence in the Discussion section: “Our results support the findings of other studies in the US that have examined similar factors within these three domains [13-36].” (Lines 349-351)

Point 18. How the authors think that their article is providing information to the literature published? Not only being the first study examining such associations could be determinant for the interest of the study.

Response 18: We agree with the reviewer that it is very important to point out how our manuscript is contributing to the literature published in this area. The Conclusions section of our manuscript summarizes why we think our findings are adding to the literature: “The results of this bi-national study highlight some of the differences and similarities in various psychosocial, behavioral, and socio-environmental factors among a multi-ethnic sample of healthy-weight, overweight, and obese youths. We hope that our findings help to demonstrate the importance of considering a wide range of risk and protective factors for obesity among adolescents. Intervention programs should use an integrated approach that addresses several of these factors to help reduce the alarmingly high rates of obesity among youth in the US and Mexico. More research is needed on how these factors may interact with each other to cause obesity since many are interrelated. Our study paves the way for future studies to focus on adopting a multi-faceted approach to identify and address important risk factors for obesity among youth.”  (Lines 439-447)

Point 19. Line 311- Only to the variation of the study design or assessment of depression are the differences? I would suggest also to the characteristics of the study sample

Response 19: As suggested by the reviewer, we have changed the sentence: “These mixed results could be attributable to variations in study design, assessment of depression [54], or due to the characteristics of the study sample.” (Lines 366-367)

Point 20. Did the meta-analysis provide any information or rationale of why overweight-obese females are more depressed? Could be something physiological in females that not occur in males?

Response 20: Unfortunately, the meta-analysis study on depression and obesity does not provide an explanation for why obese females were more likely to be depressed. Please see the response below to Point 21, for a possible physiological explanation.

Point 21. 318-319. Why do you think that girls are more likely to perceive themselves as overweight or obese? Maybe they are more strict with theirselves?

Response 21: We appreciate the reviewer’s comment about why girls are more likely to perceive themselves as overweight or obese than males, we have included some statements and a reference about physiological differences in own body perception by gender. “A recent study investigated brain activation using functional magnetic resonance imaging during a Body Perception task in healthy males and females. They found that images of their own bodies were more salient for the female participants and concluded that females may be more vulnerable than males to conditions involving own body perception [55]. (Lines 372-376).”

Point 22. Based on the interaction analyses, we will delete or justify why some associatons are occurred in girls and not in boys or viceversa.

Response 22: As indicated above, we think it is very important to present the results of our analyses stratified by sex. The final paragraph of the Results section reports the differences that were observed by sex: “Overweight or obese boys are more likely to report dissatisfaction with their body image (OR= 1.81 and OR= 3.21, respectively) than girls (OR= 1.59 and OR= 2.78, respectively). However, the presence of depressive symptoms is significantly greater among overweight and obese females (OR= 1.14 and OR= 1.16, respectively), but not among males. Girls are also more likely to perceive themselves as overweight or obese and “feel fat” (OR= 8.91 and OR= 34.28, respectively), than boys (OR= 7.14 and OR= 32.28, respectively). Obese females are significantly less likely to be physically active (OR= 0.72) and eat breakfast than healthy-weight females (OR= 0.40), but this association was not found to be significant among males. Overweight or obese males are more likely to engage in weight control behaviors (OR= 13.77 and OR= 12.69, respectively) than obese females (OR= 8.02), especially exercise (OR= 2.67 and OR= 2.59, respectively) and eating less/few calories/low fat foods (OR= 3.40 and OR= 2.93, respectively). However, obese girls are significantly more likely to consume diet pills, powders or liquids (OR= 9.59) than boys. (Table 5) All the aforementioned results had a p-value of <0.05, which was considered statistically significant.” As indicated in the Limitations paragraph of the Discussion section, this was an exploratory study, so we were not able to determine the reason for these differences by sex as part of this study. “The information provided by the study participants was of a quantitative nature, so we were unable to determine the reason for some of the differences observed by sex or country of origin. Future studies should collect more qualitative data to investigate these differences.” (Lines 421-424)

Point 23. I would discuss more the findings obtained along the discussion section, why they are agree or not with other studies? What could be the justification of the findings?

Response 23: We respectfully disagree with the reviewer on this comment. We feel that we have adequately compared our results to those of 27 other published studies in the literature. As indicated in the previous response, this was an exploratory study, so we were not able to justify or determine the reason for most of the results of this study. “The information provided by the study participants was of a quantitative nature, so we were unable to determine the reason for some of the differences observed by sex or country of origin. Future studies should collect more qualitative data to investigate these differences.” (Lines 421-424)

Point 24. In limitations, author should state also the questionnaire and self-reported measures as limitation. Also the differences in number of the sample size (Mexico much lower than usa)

Response 24: We appreciate the suggestion and have modified the Limitations paragraph of the Discussion section to address these issues: “Additionally, this is an exploratory study with a limited sample size for the participants in Mexico. Future studies should be conducted with a larger sample size that will allow for a higher significance threshold to be set for individual comparisons to compensate for the number of inferences being made. Other limitations include the specific measures that were collected using a self-reported questionnaire, a lack of validated measures, and the fact that some of the behavioral and socio-environmental indices, e.g. “Healthy Lifestyle Priorities,” “Physically Active,” “Mother/Father Healthy Values,” or “Home Availability of Healthy Foods,” were created based on a limited number of variables and should be interpreted as preliminary findings. The information provided by the study participants was of a quantitative nature, so we were unable to determine the reason for some of the differences observed by sex or country of origin. Future studies should collect more qualitative data to investigate these differences.” (Lines 414-424)

Round 2

Reviewer 3 Report

The authors have addressed all the issues stated from reviewers 1 and 2 and consequently, the manuscript has improve a lot. So, I have no further comments or suggestions for this paper.

Author Response

Thank you very much for your advice.